# The Role of Microglia in the Development of Neurodegenerative Diseases

**DOI:** 10.3390/biomedicines9101449

**Published:** 2021-10-12

**Authors:** Jae-Won Lee, Wanjoo Chun, Hee Jae Lee, Seong-Man Kim, Jae-Hong Min, Doo-Young Kim, Mun-Ock Kim, Hyung Won Ryu, Su Ui Lee

**Affiliations:** 1Natural Medicine Research Center, Korea Research Institute of Bioscience and Biotechnology (KRIBB), Cheongju 28116, Korea; ksm2906@kribb.re.kr (S.-M.K.); minjh1710@naver.com (J.-H.M.); rose73@kribb.re.kr (D.-Y.K.); 2Department of Pharmacology, College of Medicine, Kangwon National University, Chuncheon 24341, Korea; wchun@kangwon.ac.kr (W.C.); heejaelee@kangwon.ac.kr (H.J.L.)

**Keywords:** microglial activation, inflammatory molecules, neuronal damage, neurodegenerative diseases

## Abstract

Microglia play an important role in the maintenance and neuroprotection of the central nervous system (CNS) by removing pathogens, damaged neurons, and plaques. Recent observations emphasize that the promotion and development of neurodegenerative diseases (NDs) are closely related to microglial activation. In this review, we summarize the contribution of microglial activation and its associated mechanisms in NDs, such as epilepsy, Alzheimer’s disease (AD), Parkinson’s disease (PD), and Huntington’s disease (HD), based on recent observations. This review also briefly introduces experimental animal models of epilepsy, AD, PD, and HD. Thus, this review provides a better understanding of microglial functions in the development of NDs, suggesting that microglial targeting could be an effective therapeutic strategy for these diseases.

## 1. Introduction

Microglia are the patrolling immune cells in the brain. They respond to infection, closely interact with astrocytes and neurons [1], promote synaptic pruning and formation [2], and play a pivotal role in the maintenance of central nervous system (CNS) homeostasis [3]. Depending on various stimuli, microglia are activated and can be polarized into M1 and M2 phenotypes, displaying separate cell surfaces, releasing different molecules and exerting distinct functions (Figure 1) [4]. Recent discoveries through microarray and single-cell RNA sequencing reveal unique microglia known as disease-associated microglia (DAM) that express surface markers such as TERM2 and homeostatic genes, including IRF1, LXRβ, and CEBPα [5,6].

IFN-γ, TNF-α, and bacterial lipopolysaccharide (LPS) have been known to affect microglia polarization into the M1 phenotype (the classic pro-inflammatory type) [7]. In this condition, M1 microglia express various receptors and channel proteins on their surface and exert a pivotal role in host immune responses by producing pro-inflammatory cytokines (TNF-α, IL-6, and IL-1β), chemokines (MCP-1, RANTES, and eotaxin), iNOS, COX-2 and ROS [8,9,10,11]. Through this process, M1 microglia kill foreign invaders, remove cellular debris through phagocytosis, and affect T cells [12]. Microglia polarization towards the M2 phenotype (the alternative protective type) is normally initiated by the stimulation of IL-4 or IL-13, which are secreted by Th2 cells [10]. M2 microglia produce anti-inflammatory cytokines (arginase-1, IL-10 and TGF-β), chemokines (TARC and MDC), and growth factors (BDNF, VEGF, PDGF and IGF-1) that suppress inflammatory responses and contribute to tissue repair and remodeling [13,14]. Thus, this microglial activation eventually triggers neurodegeneration by inducing inflammatory responses or exerts neuroprotective effects by suppressing inflammatory responses and by recovering impairment [15].

Cumulative results show that microglia closely interact with neurons and influence the production of neurotrophic factors such as IGF-1, the neurogenesis process and synapse formation during the development of the brain [16,17,18]. Thus, the important role of microglia in immune processes and brain homeostasis is emphasized. However, the hyper-activation and dysregulation of microglia have been known to induce neurotoxic consequences. Recent observations showed that microglial activation and its associated molecules are associated with the promotion of neurodegenerative disorders, such as epilepsy, Alzheimer’s disease (AD), Parkinson’s disease (PD), and Huntington’s disease (HD). Thus, these results reflect that the modulation of microglial activation may be an effective approach for treating neurodegenerative disorders. In the present review, we describe how microglia may contribute to the development of epilepsy, AD, PD, and HD based on recent results. In addition, this review summarizes therapeutic attempts to target proteins involved in the regulation of microglia (Table 1), highlighting the importance of the regulation of microglial activation.

## 2. Microglia in Epilepsy

Neurotransmitters such as glutamate lead to excitatory currents in neurons as well as neuronal tissue damage [37]. The inhibitory neurotransmitter GABA exerts an inhibitory effect on the excitability of the CNS [38]. Epilepsy is a severe neurodegenerative disease (ND) accompanied by acute epileptic seizures, and the imbalance of synaptic excitation (E)/inhibition (I) is recognized as one of the major causes of epilepsy [39]. It has recently been suggested that glial cells (e.g., microglia) are closely associated with an imbalance of synaptic E/I and synaptic pruning in epilepsy [2]. Microglia express a variety of cell surface molecules (Iba-1), receptors (CX3CR1, GLP-1R, and TREM2), channels (TRPV4), pro-inflammatory cytokines (TNF-α, IL-1β, and IL-6)/mediators (iNOS and COX-2), and ROS [21,40,41,42,43,44]. The upregulation of ROS is related to the activation of microglia and the expression of TNF-α, IL-1β, and IL-6 [45]. Microglia-derived pro-inflammatory cytokines can induce microglial activation, neuroinflammation and ROS production, resulting in the promotion of neuronal excitability and epileptogenesis [46,47]. In epilepsy, the activation of NF-κB in microglia has been known to cause neuroinflammation by inducing pro-inflammatory cytokines and chemokines (TNF-α, IL-1β, MIP-1α, and MCP-1) [48,49]. 

The combination of the microglia-expressed fractalkine receptor CX3CR1 and the neuron-expressed ligand CX3CL1 reflects the interaction of microglia and neurons, mediates synaptic pruning and plasticity, and affects neurogenesis in the hippocampus [2,50,51,52]. Previously, Xu et al. reported that the elevation of CX3CL1 expression was confirmed in the neocortex of patients with temporal lobe epilepsy (TLE), and this elevation was also prominently shown in the hippocampus of TLE patients [53]. Eyo et al. confirmed that CX3CL1 treatment leads to increases in microglia–neuron interactions, and IL-1β plays an important role in the CX3CR-1-dependent interaction of microglia and neurons in studies of epilepsy [54]. The administration of CX3CR1 antibodies relieved seizure-induced microglial activation and neurodegeneration [21].

GLP-1R is expressed in neurons, astrocytes and microglia [54,55]. The administration of the GLP-1R agonist exendin-4 suppressed the LPS-induced increase in TNF-α/IL-1β mRNA and NF-κB activation in microglia [43]. Recent results showed that increased levels of GFAP and Iba-1-positive cells and increased TNF-α, IL-β, and IL-6 secretion were ameliorated by administration of a GLP-1R agonist in the hippocampus in a spinal nerve ligation (SNL) rat model [56]. Recent observations also showed that GLP-1R expression was decreased in the cortices of TLE patients and in the cortex/hippocampus of rats with pentylenetetrazol (PTZ)-induced TLE [19]. In that study, the application of the GLI-1R agonist liraglutide upregulated GLP-1R expression in the cortex/hippocampus of TLE rats and ameliorated epileptic seizures. In addition, liraglutide resulted in increases in GABA_A_R2/3 expression in the hippocampus of TLE rats [19]. Thus, these results indicate that targeting GLP-1R in microglia could be a valuable antiepileptic approach.

TRPV4 channels are expressed in neurons and glial cells such as astrocytes and microglia [20,40,57]. Microglia-derived neuroinflammatory cytokines, such as TNF-α and IL-1β, are related to brain injury, and the expression of these cytokines is derived from NF-κB activation [58,59]. Shi et al. have shown that the activation of TRPV4 promotes neuroinflammation and neuronal impairment by enhancing the levels of TNF-α/IL-1β and NF-κB activation in both in vitro and in vivo [60]. A recent in vitro study showed that TRPV4 suppression induced by its antagonist resulted in the inhibition of NF-κB activation and TNF-α/IL-1β production in LPS-stimulated microglia [20]. Furthermore, another recent in vivo study demonstrated that the activation of TRPV4 led to increases in microglial activation and Iba-1 expression; TNF-α, IL-1β and IL-6 expression; NLRP3, ASC, and caspase-1 expression; and neurotoxicity in experimental animal models of TLE [40]. In that study, these increased levels were suppressed by a specific TRPV4 antagonist, HC-067047. These results indicate that TRPV4 stimulation in microglia induces neuronal injury, and the suppression of TRPV4 activation ameliorates neuronal impairment by downregulating NF-κB activation and inflammatory cytokines, suggesting that TRPV4 channels in microglia could be a therapeutic target in epilepsy. 

### Experimental Animal Models of Epilepsy

The excitotoxic glutamate analog kainic acid (KA) and the GABA-A receptor antagonist pentylenetetrazol (PTZ) cause seizures and provoke inflammatory responses, oxidative stress, and neuronal damage, and have been widely used to induce status epilepticus via various routes of administration, such as intracerebroventricular (i.c.v.), intra-amygdala, intraperitoneal (i.p.), intranasal, and subcutaneous [61,62,63,64,65,66,67,68,69,70,71]. 

In rodent models of KA- or PTZ-induced epilepsy, increased levels of seizure behavior, microglial activation markers (Iba-1, TNF-α, IL-1β, IL-6, iNOS, and COX-2), NF-κB activation, apoptosis-related proteins (Bax and caspase-3/8/9) and ROS were confirmed in the hippocampus [48,61,72,73,74,75,76]. Furthermore, decreased levels of NeuN and Bcl-2 expression and AKT and CREB phosphorylation were confirmed in the hippocampus or cortices of rodents [71,77,78]. A significant increase in the interaction of microglia and neurons was confirmed by KA administration [54]. Thus, the use of KA or PTZ has been reasonably validated in the establishment of animal models of epilepsy [79]. 

It was recently introduced that compounds isolated from natural products [77], antioxidants [74] and agonists [19,54,80], exerted protective effects in epileptic studies. Hyperoside from *Hypericum perforatum* L. inhibited neuronal damage through upregulating the expression of antioxidant proteins [77]. The antioxidants asiatic acid and maslinic acid exerted protective effects against KA-induced neuroinflammation and oxidative stress by regulating inflammatory molecules, apoptosis-related molecules, and ROS/oxidized glutathione (GSSG) [74]. Amentoflavone suppressed neuronal loss and activation of the NLRP3 inflammasome of the hippocampus in PTZ-induced kindling mice [68]. CX3CR1 deficiency reduced microglia–neuron interactions and CX3CL1 treatment was shown to upregulate these interactions [54]. The GLP-1R agonist liraglutide inhibited PTZ-induced seizures and increased levels of MDA and caspase-3 in a PTZ kindling rat model [80]. Liraglutide also inhibited epileptic seizures and regulated neuronal receptor expression in PTZ-induced seizures in rat [19].

## 3. Microglia in Alzheimer’s Disease (AD)

AD is known as a common cause of dementia, and impairments of memory and cognition are major symptoms of AD [81]. The major hallmarks of AD are the increase in amyloid-β precursor protein (APP), the deposition of amyloid-β (Aβ) plaques and α-synuclein (α-Syn), the activation of astrocytes/microglia, the hyperphosphorylation and aggregation of tau, neuroinflammation and the cell death of neurons [81,82,83]. BDNF/TrkB neurotrophic signaling contributes to neuronal survival, development, and synaptic plasticity, and dysregulation of the BDNF/TrkB signaling pathway underlies neurodegeneration in AD [84,85]. It has been reported that ROS elevation causes oxidative stress, the dysfunction of astrocytes, the promotion of Aβ generation, neuronal damage, cognitive decline, and dementia in AD [86,87]. Astrocyte-derived Aβ leads to neuroinflammation/neuronal damage and Aβ-induced NF-κB activation in astrocytes could exert an important role in the inflammatory response [88,89,90,91,92,93,94,95]. 

Microglial activation and its associated inflammatory molecules could be detrimental to astrocytes, neurons, and synapses in AD [96]. Researchers have reported that increases in microglial activation and proliferation have been shown nearby amyloid plaque and are related to the promotion of Aβ formation and tau pathology in AD [96,97]. NLRP3 inflammasome activation could lead to IL-1β/IL-18 release and be related to the enhancement of Aβ and tau aggregation in AD [98,99,100,101,102]. Recently, it has been reported that NLRP3 inflammasome activation in microglia promotes tau pathology in AD [103]. A recent observation from Alves et al. indicated that phosphorylated tau was located close to microglia [62]. Interactions between astrocytes and microglia could be mediated via inflammatory cytokines (TNF-α, IL-1β, and IL-6) induced by Aβ exposure [87]. Shi et al. reported that A1 astrocytic genes were activated in TE4 mice, and microglia-derived cytokines (TNF-α, IL-1α, and C1q) could lead to A1 astrocyte activation and neuronal damage [104]. As reported in a recent study by Grimaldi et al., A1 astrocytes could be triggered by activated microglia-derived IL-1β and by the accumulation of extracellular Aβ in the retina of patients with AD [89]. These observations showed the interaction of microglia and astrocytes in the progression of AD and the important role of microglia in the regulation of A1 reactive astrocytic activity in AD. 

Apolipoprotein E (APOE), a major cholesterol transporter for neurons, is generated from neurons, astrocytes, and microglia [90,93,105,106,107,108]. Glial-cell-derived APOE leads to APP transcription and Aβ secretion in neurons [109]. Shi et al. showed that the deletion of APOE4 was neuroprotective in tau-mediated neurodegeneration in tauopathy mice [104]. Recently, Shi et al. also reported that neurodegeneration with APOE dependence is derived from microglia in tauopathy mice [82]. In that study, the depletion of microglia suppressed the promotion of tau pathology, and the increase in Aβ pathology was closely associated with APOE4 relative to different APOE isoforms [82]. Furthermore, a recent review article described the importance of anti-APOE4 immunotherapies in therapeutic approaches for AD [110]. These results indicate that microglia could drive APOE-dependent neurodegeneration in AD. Thus, APOE4 is emphasized as the strongest prevalent genetic risk factor for AD, and the inhibition of microglial APOE4 could be a valuable therapeutic approach in AD.

The interaction of the microglia surface receptor TREM2 with APOE is known to modulate AD pathology [111,112]. Recently, in vitro and in vivo studies showed that TREM2 deficiency reduces Aβ uptake by microglia administered APOE4, and the interaction of Aβ with microglia could be affected by isoforms of APOE [113]. Recent in vitro results from Fuganzu et al. indicated that TREM2 overexpression inhibited TNF-α, IL-6, and IL-1β, and led to the upregulation of Arg-1, IL-10, and Ym1 in Aβ1–42-stimulated BV2 microglia [114]. In that study, TREM2 overexpression reduced the accumulation of Aβ and the expression of BACE-1, Iba-1, GFAP, TNF-α, IL-6, and IL-1β in both the cortex and hippocampus in an APP/PS1 transgenic mouse model of AD and also led to the upregulation of M2 phenotype markers (Arg-1, IL-10, and Ym1) in both the cortex and hippocampus [114]. Interestingly, these effects were accompanied by the suppression of JAK2 and STAT1 activation and the reduction of SOCS1 and SOCS3 both in vivo and in vitro. Recently, Wang et al. showed that the hTREM2 antibody AL002c acted as a TREM2 agonist, promoting the activation of microglia, Aβ phagocytosis, and inhibition of filamentous plaques in a study of AD [26]. Furthermore, recent observation showed that TREM2 upregulation reduces ApoE4-associated cognitive impairment and neurodegeneration [115]. These findings provide conflicting results showing whether the response of microglial TREM2 in AD pathophysiology is protective or detrimental. Considering the results of preclinical studies which show the beneficial effect of TREM2-associated activation of microglia on tau hyperphosphorylation [22,116] and Aβ phagocytosis [32,112], TREM2 could be a valuable therapeutic target in AD. Additional findings may provide evidence for the usefulness of TREM2.

It is known that dipeptidyl peptidase-4 (DPP-4) leads to the inactivation of GLP-1; thus, DPP-4 inhibitors lead to increased levels of GLP-1 and have been usefully adapted in various AD animal models [24,25]. GLP-1R has been proposed as a target in AD therapy [117,118]. The findings of a recent in vitro study by Park et al. showed that the GLP-1R agonist NLY01 inhibited the increased mRNA levels of TNF-α, IL-1α, IL-1β, IL-6, and C1q in Aβ-stimulated microglia and also suppressed Aβ-induced MAP2, Bcl-2, and BDNF reduction and neuronal cell death [119]. In addition, NLY01 suppressed the increased levels of TNF-α, IL-1β, IL-6, C1q and IFN-γ in 3xTg-AD mice [119]. In both 5xFAD and 3xTg-AD transgenic mouse models of AD, NLY01 suppressed microglia-mediated reactive astrocyte conversion and contributed to improvements in spatial learning and memory [119]. Collectively, these results emphasize the importance of microglia GLP-1R in AD progression, suggesting that targeting the GLP-1R of microglia may be an effective therapeutic approach in AD (Figure 2). 

### Experimental Animal Models of AD

Researchers have tried to improve the progression of AD in preclinical studies, and AD animal models have exhibited AD pathology, including Aβ accumulation. With the necessity of better AD animal models, new animal models have been produced and developed. APOE4 is implicated in the deposition of Aβ [110] and increases in neurodegenerative markers. It is also more strongly associated with cognitive decline compared to APOE2 and APOE3 in APOE-Tg mouse models [23,120,121,122]. 5xFAD and 3xTg-AD mice exhibited an overproduction of APP/Aβ and the development of cognitive deficits [119].

The APOE antagonist 6KapoEp decreased Aβ and tau pathologies in 3xTg-AD and 5xFAD mice [27]. GLP-1R agonists such as exendin-4 exert protective effects in experimental models of AD [118]. Furthermore, the GLP-1R agonist NLY01 improved AD symptoms by regulating the microglia-mediated conversion of reactive astrocytes and decreased memory deficits in 5xFAD and 3xTg-AD mice, indicating that NLY01 could be a viable therapy in AD [119].

Previously, it has been reported that the DPP-4 inhibitor saxagliptin exerted a neuroprotective effect by ameliorating memory and learning deficits, upregulating hippocampus GLP-1 and downregulating hippocampus Aβ, p-tau, TNF-α, and IL-1β in streptozotocin-induced AD rats [28]. A recent observation showed that the DPP-4 inhibitor vildagliptin ameliorated neurodegeneration by upregulating BCL-2 and Klotho, and suppressing TNF-α, FOXO1, Bax, and caspase-3 in HFHS diet/AlCl3-induced AD rats [123]. Kosaraju et al. reported that the oral administration of the DPP-4 inhibitor linagliptin inhibited increases in the levels of Aβ, tau phosphorylation, neuroinflammation, and cognitive deficits in 3xTg-AD mice [24].

A recent research paper indicated the usefulness of a TrkB agonistic antibody (AS86) in an APP/PS1 mouse model by showing its ability to modify spatial cognition deficiency [124]. It is well known that the TrkB agonist 7,8-dihydroxyflavone (7,8-DHF) inhibits memory deficits and β-secretase enzyme and increases p-TrkB in 5xFAD transgenic AD mice [125]. The recent results from Chen et al. showed that a prodrug of 7,8-DHF, R13, led to the activation of TrkB and downstream signaling (AKT and ERK); the alleviation of Aβ deposition; and TNF-α, IL-6, and IL-1β production in 5xFAD mice [126]. In that study, R13 also exerted protective effects against synaptic loss and spatial learning and memory impairment, indicating that R13 may be a neuroprotective agent for AD. 

There have been various reports confirming the protective effects of antioxidants in AD preclinical studies [86,127]. A recent AD study showed that linalool, a natural compound with antioxidant effects, inhibited increased levels of oxidative stress and GFAP in rats with Aβ injection [128].

## 4. Microglia in Parkinson’s Disease (PD)

PD is a movement disorder associated with cognition and memory problems and is the second most common ND after AD [129]. The impairment of dopamine (DA) neurons, inhibition of tyrosine hydroxylase (TH), lower striatal dopamine transporter (DAT) availability in substantia nigra (SN), the accumulation of Lewy bodies arise from α-Syn accumulation, and the activation of A1 neurotoxic astrocytes are pathological features of PD [117,130,131,132]. 

Astrocytes exert important roles in PD pathology through functional changes [94]. Ca^2^^+^ signals in astrocytes are regulated by DA [133]. The loss of DA neurons could lead to changes in the Ca^2^^+^ homeostasis of astrocytes, and Ca^2+^ imbalance could lead to the production of toxic molecules and cell death in PD [134]. A recent study showed that increased levels of α-Syn expression were confirmed in astrocytes from PD through immunocytochemistry (ICC) and qRT-PCR assays [130]. In that study, PD astrocytes upregulated the secretion of IL-6 and RANTES and the mRNA expression of LCN2 and GFAP against inflammatory stimulation (e.g., TNF-α or IL-1β exposure). In addition, that study also showed increased Ca^2+^ levels in the ER of PD astrocytes [130]. Accumulating results show that astrocyte-derived glia maturation factor (GMF) causes the activation of NF-κB and the secretion of GM-CSF, and the increased levels of GM-CSF may lead to the activation of microglia and the secretion of inflammatory molecules such as TNF-α, IL-1β and MIP-1 β [135,136,137]. A recent study revealed that GMF deficiency suppressed the expression of NLRP3 in both astrocytes and microglia in SN of MPTP-induced PD mice [138]. In that study, GMF deficiency also inhibited the damage of DA neurons by inhibiting the expression of ASC and caspase-1 [138]. 

Neuroinflammation leads to a loss of DA neurons in PD and microglia-derived TNF-α, IL-1β, and IL-6, which leads to neuroinflammation in PD [139]. NF-κB activation has been detected in the midbrain of PD patients and in MPTP mouse models of PD, and the selective suppression of NF-κB ameliorated neuroinflammation in MPTP-induced PD mice [140]. The information from a recent review article emphasizes the contribution of microglia in the neuroinflammation and phagocytosis of α-Syn in PD [141]. Liddelow et al. reported that microglia-derived TNF-α, IL-1β, and C1q could induce neurotoxic A1 reactive astrocytes in ND including PD [142]. Thus, the regulation of microglia-derived A1 astrocyte conversion could be an important strategy in PD. 

In PD, APOE is a known risk gene and is closely related to the promotion of α-Syn and Lewy bodies; therefore, it is thought be an important therapeutic target in PD [143,144]. In these studies, the increase in PD pathological hallmarks, such as α-Syn, behavioral impairment, neuron loss, and astrogliosis, was confirmed in α-Syn-APOE4 mice. Increased levels of the APOE receptor TREM2 were identified along with the upregulation of M1 macrophage cytokines (TNF-α, IL-1β and IL-6) and the M2 macrophage gene Arg-1 in the midbrain of MPTP-induced PD mice [145]. Microglia are known to express TREM2 and produce APOE [112]. TREM2 suppression in BV2 microglia promoted the inflammatory responses of M1 microglia, and the upregulation of TREM2 led to the promotion of M2 polarization and the alleviation of microglial inflammatory responses, indicating that TREM2 in microglia could lead to the transformation of M1 microglia into the M2 phenotype, leading to anti-inflammatory effects in PD [145]. Considering that that the inhibition of M1 microglia activation through the transformation into the M2 phenotype microglia could be a potential treatment in PD [29], TREM2 in microglia may be an important factor in modifying microglial phenotype. 

Recent evidence suggests that DPP-4 inhibitors have beneficial effects in PD [30,146,147]. In clinical trials, DPP-4 inhibitors preserved striatal DAT availability, which is known to be associated with anxiety or depression in PD patients [33,146,148]. In in vivo studies of PD, DPP-4 inhibitors, such as saxagliptin and linagliptin, relieved neuroinflammation and neuron damage in PD animal models [149]. 

Recently, the therapeutic use of GLP-1R agonists has been suggested in PD because of this receptor’s important role in the prevention of DA neuron loss and microglia-mediated neurotoxic A1 reactive astrocytes [117]. In that study, the GLP-1R agonist NLY01 effectively suppressed the α-synuclein preformed fibril (α-Syn PFF)-induced microglial activation marker Iba-1 and the secretion of TNF-α, IL-1α, and C1q, and exerted inhibitory effects on the nuclear translocation of NF-κB in α-Syn PFF-stimulated microglia [117]. A recent observation from Wang et al. showed that the GLP-1R agonist PT320 ameliorated the progression of PD through improvements in behavior and DA midbrain function in a progressive PD mouse model [150]. Therefore, these results suggest that the activation of GLP-1R in microglia could suppress A1 neurotoxic astrocytes and ameliorate progression in PD (Figure 2). 

### Experimental Animal Models of PD

1-Methyl-4-phenyl-1,2,3,6-tetrahydropyridine (MPTP), which is known as a potent neurotoxin, has been used for establishing PD in animal models, and it led to astrocyte/microglia activation, pro-inflammatory cytokine release (TNF-α, IL-1β and IL-6) and NF-κB activation in brain, as well as behavioral impairment [138,151,152]. Lee et al. reported that MPTP-derived NLRP3 inflammasome activation in microglia led to neuronal damage and IL-1 receptor antagonists exerted protective effects on MPTP-induced neuronal damage [99]. A recent observation from Shao et al. showed that TLR4 deficiency exerts protective effects in MPTP-induced PD mice by regulating motor impairment, DA neuronal damage, astrocyte/microglia activation, α-Syn and NLRP3/NF-κB activation [151]. A recent result from Song et al. showed that 2-hydroxy-4-methylbenzoic anhydride (HMA) decreased the activation of microglia and the expression of Iba-1, GFAP and COX-2 in the striatum of MPTP-induced PD mice [153]. It is well known that NBD-peptide-induced NF-κB inactivation causes the downregulation of mRNA levels of GFAP, CD11b, iNOS, TNF-α, and IL-1β in the midbrain of MPTP-induced PD mice [140]. 

A previous study demonstrated that the intrastriatal administration of α-Syn in nontransgenic mice led to neurodegeneration seen in PD, such as damage to DA neurons and motor deficits [31]. hA53T α-Syn transgenic mice have been revealed to have increased levels of α-Syn, neuronal loss and motor dysfunction [34,117,154]. Therefore, this mouse model has been used for PD studies. As introduced in AD studies, GLP-1R agonists have been suggested as potential neuroprotective agents for PD. Recent results showed that GLP-1R agonists such as NLY01 inhibited the impairment of dopaminergic neurons in PD mice induced by the intrastriatal injection of α-Syn PFF [117]. In that study, NLY01 also ameliorated the behavioral deficits in human A53T α-Syn (hA53T) transgenic mice [117]. Furthermore, NLY01 inhibited microglial activation, pro-inflammatory molecules, and NF-κB activation against α-Syn PFF stimulation [117]. These results suggest that NLY01 may be useful for PD therapy. 

As proven by the protective effect of DPP-4 inhibitors in AD studies, their ameliorative effects were shown in PD studies. The DPP-4 inhibitor saxagliptin inhibited the decreased levels of DA neurons; cAMP, BDNF and BCL-2; and increased levels of NF-κB, TNF-α, iNOS, ICAM-1, and MPO in rotenone-induced PD rats [30]. In that study, saxagliptin also led to the inhibition of Cyt C and caspase-3 in rotenone-induced PD rats, indicating an antiparkinsonian effect [30]. Linagliptin also exerted ameliorative effects in MPTP-induced PD mice by suppressing TLR4, NF-κB, and TNF-α, upregulating GLP-1 and TH, and improving behavioral changes [35]. 

The regulation of BDNF/TrkB signaling is considered an important therapeutic strategy in PD [155]. The TrkB agonist 7,8-DHF protected DA neurons in both a 6-hydroxydopamine (6-OHDA)-induced PD rat model and an MPTP-induced PD mouse model [156]. Recent observations in an in vivo study showed that 7,8-DHF administration led to the activation of TrkB signaling, reductions in α-Syn and tau phosphorylation, protection of DA neurons, and improvements of behavioral deficits in a rotenone-induced rat model of PD, suggesting that 7,8-DHF may be a useful agent in PD therapy [36]. 

## 5. Microglia in Huntington’s Disease (HD)

HD is a severe autosomal-dominant genetic disorder and is caused by the repeated expansion of cytosine–adenine–guanine (CAG) in the huntingtin (Htt) gene [157]. Abnormal movement, chorea, and cognitive impairment are exhibited in patients with HD [158], and this pathology is derived from the progressive death of striatal neurons [159].

Previously, Faideau et al. reported that the expression of mutant huntingtin (mHtt) in astrocytes may be related to alterations in glutamate transport capacity and could lead to HD pathogenesis [160]. Recently, a review article from Palpagama et al. suggested that reactive astrocytes lead to neuronal damage in HD by generating inflammatory cytokines and neurotoxic molecules, including ROS and quinolinic acid [161]. In that study, the effects of activated microglia-derived inflammatory cytokines, ROS, and quinolinic acid on neuronal death in HD were also described [161]. It has previously been reported that monocytes derived from HD subjects express mHtt [162], and microglia expressing mHtt derived from YAC128 HD mice significantly upregulated the production of inflammatory cytokines, including IL-1β, compared to wildtype microglia against stimuli such as LPS [163]. Lopez-Sanchez et al. recently reported that 3-NP led to microglia-derived TNF-α and IL-1α and neurotoxic A1 astrocytes in the striatum, hippocampus, and cerebellum in rats [164]. Microglial density and phagocytosis were increased in the striatum of an R6/2 mouse model of HD [165]. Recent research also emphasized the important role of microglia in HD by showing that the depletion of microglia could ameliorate changes in the extracellular matrix and reduce the striatal volume in animal models [166]. These observations indicate that microglial activation and consequent neuroinflammation are currently recognized as key features of HD and are closely related to the development of HD. Thus, recently gathered data suggest that targeting microglia could be an effective approach in HD therapy.

Recent in vivo observation results also demonstrated the ameliorative effects of DPP-4 in HD studies [159]. Previous observations showed that exendin-4 (Ex-4), an FDA-approved antidiabetic GLP-1R agonist, exerted a protective effect in HD mice by reducing mHtt accumulation and improving motor function [149]. A recent review article described the neuroprotective role of GLP-1R stimulation in neurological disorder, including HD [167]. As proven in epilepsy, AD, and PD studies, the important role of GLP-1R is being confirmed in HD studies. Thus, targeting GLP-1R may be a valuable therapeutic approach for NDs including HD.

### Experimental Animal Models of HD

As shown in epilepsy, AD, and PD research, investigations using in vivo models are active and ongoing in HD research and could provide insights into the pathogenesis of HD and the therapeutic benefits of candidates.

The i.p. administration of 3-nitropropionic acid (3-NP) leads to striatum degeneration and neurological disturbance in rodents. Therefore, it has been used for establishing HD animal models. Sayed et al. discussed that the DPP-4 inhibitor vildagliptin, which is known to exert protective effects in AD and PD, has recently been suggested as a promising therapeutic in HD [159]. In that study, vildagliptin suppressed 3-NP-induced striatal degeneration by restoring AKT, CREB, BDNF, GABA, Srt-1, and Nrf2 and suppressing MDA, glutamate and GFAP. In line with the protective effect of the TrkB agonist 7,8-DHF in previous AD and PD studies, recent observations demonstrated its protective effects in experimental animal models of 3-NP-induced HD by promoting the activation of CREB and AKT and suppressing neuronal death [168]. 

In a recent report by Yang et al., ginsenoside Rg1 (Rg1) exhibited neuroprotective properties and behavioral defects in 3-NP-induced HD mice through inhibition of the MAPK and NF-κB pathways [169]. In that study, Rg1 exerted inhibitory effects on microglial activation and inflammatory molecules in the striatum. Furthermore, a recent study confirmed that the elimination of microglia induced by the CSF1 inhibitor PLX3397 relieved the accumulation of mHtt, the extension of astrogliosis and the loss of striatal volume in the transgenic R6/2 mouse model, which is commonly used for HD [166]. The GLP-1R agonist Ex-4 is known to exert ameliorative effects in N171-82Q transgenic HD mice [149].

## 6. Conclusions

Currently, researchers are exploring a variety of therapeutic approaches that include agonists and antagonists in the therapy of NDs. Recent observations suggest that microglia are strongly related to the progression of NDs, such as epilepsy, AD, PD, and HD. Thus, a better understanding of microglia functions could be important. Moreover, regulating microglial activation could ameliorate the development of NDs and may provide novel therapeutic strategies in the therapy of NDs. The identification of promising and shared therapeutic targets, such as DPP-4, GLP-1R, and TrκB, in different diseases, such as AD, PD, and HD, suggest its importance in therapeutic approaches to NDs. Considering the fact that there are shared targets in NDs, therapeutic approaches to regulate microglia activation are important and emphasized in NDs. Further investigation will make it possible to clarify the role of microglia in NDs and to apply potential candidates for ND therapy.

## Figures and Tables

**Figure 1 biomedicines-09-01449-f001:**
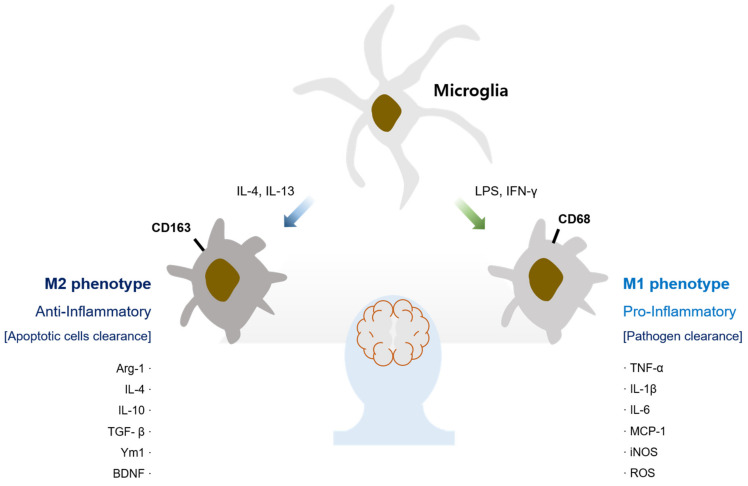
Microglial phenotypes in the brain. In response to stimuli, microglia can be classified into M1 and M2 phenotypes expressing a variety of surface markers and inflammation-associated molecules. The binding of LPS and Th1 cytokines (IFN-α and TNF-α) to receptors on microglia leads to M1 phenotype microglia, which generate pro-inflammatory molecules. Th2 cytokines (IL-4 and IL-13) promote M2 phenotype microglia, which in turn generate anti-inflammatory molecules. This process takes place in a variety of ways.

**Figure 2 biomedicines-09-01449-f002:**
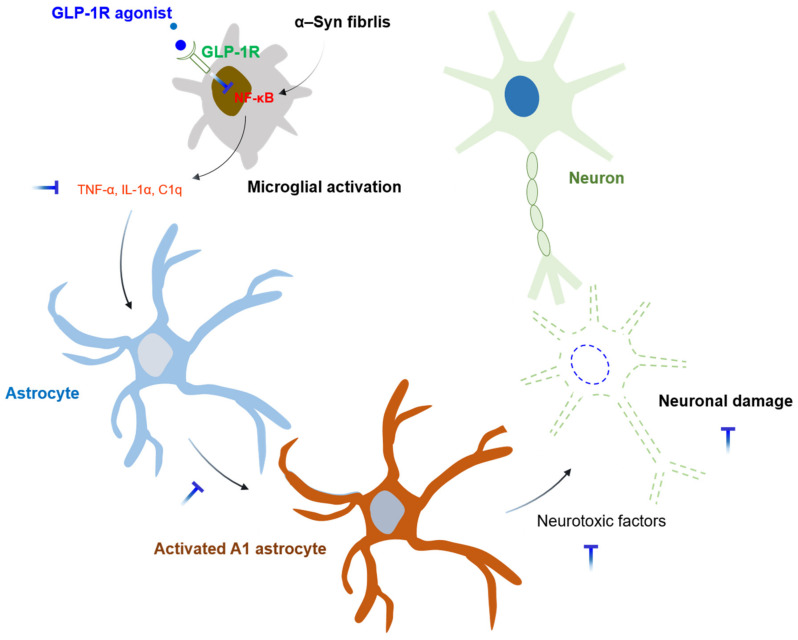
The role of microglial GLP-1R in AD and PD. The binding of GLP-1 to GLP-1R on microglia promotes the expression of pro-inflammatory cytokines and neurotoxic molecules, leading to A1 astrocyte activation and neuronal damage. GLP-1 agonists exert neuroprotective effects in AD and PD by regulating NF-κB activation and A1 astrocyte activation and suppressing neuronal damage. Therefore, GLP-1/GLP-1R signaling is highlighted in novel therapeutic strategies in AD and PD.

**Table 1 biomedicines-09-01449-t001:** Microglia-targeting for NDs.

Diseases	Targeting		Agent	Reference
Epilepsy	GLP-1R	Agonist	Liraglutide	[19]
	TRPV4	Antagonist	HC-067047	[20]
	CX3CR1	Antibody	Anti-CX3CR1 antibody	[21]
AD	DPP-4	Inhibitor	Saxagliptin	[22]
			Linagliptin	[23]
	GLP-1R	Agonist	Exendin-4	[24]
			NLY01	[25]
	TREM2	Agonist	AL002c	[26]
	TrкB	Agonist	AS86	[27]
			7,8-dihydroxyflavone	[28]
PD	DPP-4	Inhibitor	Saxagliptin	[29,30]
			Linagliptin	[31]
	GLP-1R	Agonist	NLY01	[32]
			PT320	[33]
	TrкB	Agonist	7,8-dihydroxyflavone	[34,35]
HD	DPP-4	Inhibitor	Vildagliptin	[36]
	GLP-1R	Agonist	Exendin-4	[30]
	TrкB	Agonist	7,8-dihydroxyflavone	[36]

## Data Availability

No new data were created or analyzed in this study. Data sharing is not applicable to this article.

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
