# Peer review of "The Role of Microglia in the Development of Neurodegenerative Diseases"

_biomedicines, 2021, doi:10.3390/biomedicines9101449_

Round 1
Reviewer 1 Report
The authors have described an interesting review about the role of microglia cells in several critical neurodegenerative disorders such as Alzheimer’s, Parkinson’s, Huntington’s disease and Epilepsy. The current topic is quite remarkable as more studies have pointing inflammation as a key factor for these disorders. Although the manuscript is well describe, I provide several comments that the authors should address.
Although M1 and M2 microglia is a typical classification, new methodology and terminology is now used due to bulk and single cell RNAseq findings. As such, I suggest that the author should use the term disease associated Microglia stages (DAMs) instead of M1 and M2 in the manuscript.
The authors should provide a more profound discussion and description of the critical findings about the role of APOE and TREM2 in several of the Neurodegenerative disorders such as Alzheimer. There are quite new elegant and significant recent studies showing the important role of these genetics factors in microglia homeostasis and function and disease progression.
I suggest the author to add a new section to summarize the similarities and differences on the role of microglia in these disorders. It would be quite interesting to discuss common and differential pathological pathways between these disorders.
Author Response
Reviewer 1)
Comments and Suggestions for Authors
The authors have described an interesting review about the role of microglia cells in several critical neurodegenerative disorders such as Alzheimer’s, Parkinson’s, Huntington’s disease and Epilepsy. The current topic is quite remarkable as more studies have pointing inflammation as a key factor for these disorders. Although the manuscript is well describe, I provide several comments that the authors should address.
Although M1 and M2 microglia is a typical classification, new methodology and terminology is now used due to bulk and single cell RNAseq findings. As such, I suggest that the author should use the term disease associated Microglia stages (DAMs) instead of M1 and M2 in the manuscript.
→ Thank you so much for your comments, which improve the quality of work. As your suggestion, we included sentences as below. Thank you so much for your comments, which improve the quality of work.
Recent discoveries through microarray and single-cell RNA sequencing reveal unique microglia known as disease-associated microglia (DAM) that express surface markers such as TERM2 and homeostatic genes including IRF1, LXRβ and CEBPα [5,6].
References)
[5] Glia 2019, 67, 1958-1975, doi:10.1002/glia.23678.
[6] Cell 2017, 169, 1276-1290 e1217, doi:10.1016/j.cell.2017.05.018.
The authors should provide a more profound discussion and description of the critical findings about the role of APOE and TREM2 in several of the Neurodegenerative disorders such as Alzheimer. There are quite new elegant and significant recent studies showing the important role of these genetics factors in microglia homeostasis and function and disease progression.
→ Thank you so much for your comments, which improve the quality of work. As your suggestion, we included sentences as below.
Furthermore, recent observation showed that TREM2 upregulation reduces ApoE4-associated cognitive impairment and neurodegeneration [101]. These findings provide conflicting results showing whether the response of microglial TREM2 in AD pathophysiology is protective or detrimental. Considering the results of preclinical studies, which show the beneficial effect of TREM2-associated activation of microglia on tau hyperphosphorylation [102,103] and Aβ phagocytosis [97,104], TREM2 could be a valuable therapeutic target in AD.
References)
[101] Mol Neurodegener 2020, 15, 57, doi:10.1186/s13024-020-00407-2.
[102] Nat Neurosci 2019, 22, 1217-1222, doi:10.1038/s41593-019-0433-0.
[103] Inflammation 2018, 41, 811-823, doi:10.1007/s10753-018-0735-5.
[104] Nat Neurosci 2019, 22, 191-204, doi:10.1038/s41593-018-0296-9.
I suggest the author to add a new section to summarize the similarities and differences on the role of microglia in these disorders. It would be quite interesting to discuss common and differential pathological pathways between these disorders.
→ Thank you so much for your comments, which improve the quality of work. As your suggestion, we included Table.1, which reflect the target of microglia in neurodegenerative disorders, such as Epilepsy, AD, PD and HD. Thank you again for providing contribution to improving the quality of work.
Table 1. Microglia-targeting for NDs
|
Diseases |
Targeting |
|
Agent |
Reference |
|
Epilepsy |
GLP-1R |
Agonist |
Liraglutide |
[40] |
|
TRPV4 |
Antagonist |
HC-067047 |
[42] |
|
|
CX3CR1 |
Antibody |
Anti-CX3CR1 antibody |
[22] |
|
|
AD |
DPP-4 |
Inhibitor |
Saxagliptin |
[102] |
|
Linagliptin |
[112] |
|||
|
GLP-1R |
Agonist |
Exendin-4 |
[105] |
|
|
NLY01 |
[106] |
|||
|
TREM2 |
Agonist |
AL002c |
[100] |
|
|
TrкB |
Agonist |
AS86 |
[114] |
|
|
7,8-dihydroxyflavone |
[115] |
|||
|
PD |
DPP-4 |
Inhibitor |
Saxagliptin |
[139][142] |
|
Linagliptin |
[150] |
|||
|
GLP-1R |
Agonist |
NLY01 |
[104] |
|
|
PT320 |
[143] |
|||
|
TrкB |
Agonist |
7,8-dihydroxyflavone |
[152][153] |
|
|
HD |
DPP-4 |
Inhibitor |
Vildagliptin |
[156] |
|
GLP-1R |
Agonist |
Exendin-4 |
[142] |
|
|
TrкB |
Agonist |
7,8-dihydroxyflavone |
[156] |
We also include the sentences in Conclusion as below.
The identification of promising and shared therapeutic targets, such as DPP-4, GLP-1R and TrκB in the different diseases, such as AD, PD and HD suggest its importance in therapeutic approach of NDs. Considering that there are shared targets in NDs, therapeutic approach to regulate microglia activation is important and emphasized in NDs.
Reviewer 2 Report
Lee and coworkers present a review of the Role of Microglia in the Development of Neurodegenerative Diseases.
In general is a well documented piece of work but in general when you read it seems to be a little bit chaotic. The reviewer thinks that the authors should try to order the ideas and establish a logic read flow.
Going in more in detail:
1 - Introduction is too much short. They should better explain the importance of the conection between microglia and NDs. They can add for instance a paragraph summarizing the therapeutic attempts to target proteins involved in the regulation of microglia.
In addition to that, the authors state "In the present review, we describe how microglia may contribute to the development of epilepsy, AD, PD and HD based on recent results ",but in the opinion fo the reviewer this is not the main idea a reader gets from the text. In the reviewer opinion the main objective of the text seems to be to explain approaches that can modulate microglia via targeting proteins that are related to it. They explain how these targets modulate microglia but there are more detailed explanations about the therapeutic approaches per target than of its relation with microglia modulation.
2 - The conclusions are too much short. One would expect a better discussion. For instance having into account that there are shared targets between the diseases, they can explain what does it means, how this can affect the possible possible therapeutic efforts against these targets, etc.
3 - The authors try to first list possible therapeutic approaches reported in the literature involving targets and molecules, given the cases, involving microglia and one of the diseases. Then they explain experimental animal models that have been used to study the relation between microglia and each disease.
However in some cases there more therapeutic approaches like TRPV4 channels in epilepsy for that an animal model is not explained after. So the reviewer believes that the authors should:
1 - Explain always how the relation between targets and microglia have been found and also when molecules are involved at which level this interaction has been found. In vitro, in vivo, In silico, etc (In some cases it is indicated like "...A recent in vitro study showed that TRPV4 suppression induced by its antagonist resulted in the inhibition ...", but not always)
2 - If some target or target-molecule relation has been studied with an animal model always report it (in case there is some of them not reported).
4 - Related with the previous point, the reviewer does not understand the importance of the mentioned animal models as some of the models were not specifically designed to study microglia. They are general animal models. The authors can be sure that they can be use to study the relation of mciroglia-disease specifically? If this is not the case, better to list/explain related animal models with the targets or targets-molecules approaches descried per diseases, it would be interesting to list/explain all the possible experimental approaches in vitro, in vivo, etc that exists. This would make the review much more logic and complete. But this is up to the authors
5 - In addition to the suggested addition in the conclusion,it would be interesting to have a paragraph analyzing/ hypothesizing why some targets are found as "key" for different diseases, how this can be related with the role of microglia in the different diseases, its relation with possible therapeutic approaches, etc.
6 - It would be also nice to have some lines explaining if some of the listed targets or molecules have been identified in an attempt to target modulate microglia to treat NDs.
Author Response
Reviewer 2)
Comments and Suggestions for Authors
Lee and coworkers present a review of the Role of Microglia in the Development of Neurodegenerative Diseases.
In general is a well documented piece of work but in general when you read it seems to be a little bit chaotic. The reviewer thinks that the authors should try to order the ideas and establish a logic read flow.
Going in more in detail:
1 - Introduction is too much short. They should better explain the importance of the conection between microglia and NDs. They can add for instance a paragraph summarizing the therapeutic attempts to target proteins involved in the regulation of microglia.
In addition to that, the authors state "In the present review, we describe how microglia may contribute to the development of epilepsy, AD, PD and HD based on recent results ",but in the opinion fo the reviewer this is not the main idea a reader gets from the text. In the reviewer opinion the main objective of the text seems to be to explain approaches that can modulate microglia via targeting proteins that are related to it. They explain how these targets modulate microglia but there are more detailed explanations about the therapeutic approaches per target than of its relation with microglia modulation.
→ Thank you so much for your comments, which improve the quality of work. We include the new sentences as below.
Recent discoveries through microarray and single-cell RNA sequencing reveal unique microglia known as disease-associated microglia (DAM) that express surface markers such as TERM2 and homeostatic genes including IRF1, LXRβ and CEBPα [5,6].
As your suggestion, we included sentences in introduction as below. Thank you so much for your comments, which improve the quality of work.
In addition, this review summarizes therapeutic attempts to target proteins involved in the regulation of microglia (Table 1), highlighting the importance of regulation of microglial activation.
2 - The conclusions are too much short. One would expect a better discussion. For instance having into account that there are shared targets between the diseases, they can explain what does it means, how this can affect the possible therapeutic efforts against these targets, etc.
→ Thank you so much for your comments, which improve the quality of work. As your suggestion, we included sentences as below.
The identification of promising and shared therapeutic targets, such as DPP-4, GLP-1R and TrκB in the different diseases, such as AD, PD and HD suggest its importance in therapeutic approach of NDs. Considering that there are shared targets in NDs, therapeutic approach to regulate microglia activation is important and emphasized in NDs.
3 - The authors try to first list possible therapeutic approaches reported in the literature involving targets and molecules, given the cases, involving microglia and one of the diseases. Then they explain experimental animal models that have been used to study the relation between microglia and each disease.
However in some cases there more therapeutic approaches like TRPV4 channels in epilepsy for that an animal model is not explained after. So the reviewer believes that the authors should:
1 - Explain always how the relation between targets and microglia have been found and also when molecules are involved at which level this interaction has been found. In vitro, in vivo, In silico, etc (In some cases it is indicated like "...A recent in vitro study showed that TRPV4 suppression induced by its antagonist resulted in the inhibition ...", but not always)
→ Thank you so much for your comments, which improve the quality of work. As your suggestion, we tried to organize in vitro and in vivo order as much as possible.
2 - If some target or target-molecule relation has been studied with an animal model always report it (in case there is some of them not reported).
→ Thank you so much for your comments.
4 - Related with the previous point, the reviewer does not understand the importance of the mentioned animal models as some of the models were not specifically designed to study microglia. They are general animal models. The authors can be sure that they can be use to study the relation of mciroglia-disease specifically? If this is not the case, better to list/explain related animal models with the targets or targets-molecules approaches descried per diseases, it would be interesting to list/explain all the possible experimental approaches in vitro, in vivo, etc that exists. This would make the review much more logic and complete. But this is up to the authors.
→ Thank you so much for your comments, which improve the quality of work. As your suggestion, we included sentences as below.
- Microglia in Epilepsy
Recent results showed that increased levels of GFAP and Iba-1-positive cells and increased TNF-α, IL-β and IL-6 secretion were ameliorated by administration of a GLP-1R agonist in the hippocampus in a spinal nerve ligation (SNL) rat model [39]. The administration of the GLP-1R agonist exendin-4 suppressed the LPS-induced increase in TNF-α/IL-1β mRNA and NF-κB activation in microglia [26].
→ The administration of the GLP-1R agonist exendin-4 suppressed the LPS-induced increase in TNF-α/IL-1β mRNA and NF-κB activation in microglia [26]. Recent results showed that increased levels of GFAP and Iba-1-positive cells and increased TNF-α, IL-β and IL-6 secretion were ameliorated by administration of a GLP-1R agonist in the hippocampus in a spinal nerve ligation (SNL) rat model [39].
Shi et al. have shown that the activation of TRPV4 promotes neuroinflammation and neuronal impairment by enhancing the levels of TNF-α/ IL-1β and NF-κB activation [45]
→ Shi et al. have shown that the activation of TRPV4 promotes neuroinflammation and neuronal impairment by enhancing the levels of TNF-α/ IL-1β and NF-κB activation in both in vitro and in vivo [45]
- Microglia in Alzheimer’s Disease (AD)
The interaction of the microglia surface receptor TREM2 with APOE is known to modulate AD pathology [96,97]. A recent study showed that the interaction of Aβ with microglia could be affected by isoforms of APOE, and TREM2 deficiency reduced the uptake of Aβ by microglia administered APOE4 [98].
→ The interaction of the microglia surface receptor TREM2 with APOE is known to modulate AD pathology [96,97]. Recently, in vitro and in vivo studies showed that TREM2 deficiency reduces Aβ uptake by microglia administered APOE4, and the interaction of Aβ with microglia could be affected by isoforms of APOE [98].
Recent results from Fuganzu et al. indicated that TREM2 overexpression reduced the accumulation of Aβ and the expression of BACE-1, Iba-1, GFAP, TNF-α, IL-6 and IL-1β in both cortex and hippocampus in an APP/PS1 transgenic mouse model of AD [99]. In that study, TREM2 overexpression also led to the upregulation of M2 phenotype markers (Arg-1, IL-10 and Ym1) in both the cortex and hippocampus [99]. In an in vitro study, TREM2 overexpression inhibited TNF-α, IL-6 and IL-1β, and led to the upregulation of Arg-1, IL-10 and Ym1 in Aβ1–42-stimulated BV2 microglia [99]. Interestingly, these effects were accompanied by the suppression of JAK2 and STAT1 activation and the reduction of SOCS1 and SOCS3 both in vivo and in vitro.
→ Recent in vitro results from Fuganzu et al. indicated that TREM2 overexpression inhibited TNF-α, IL-6 and IL-1β, and led to the upregulation of Arg-1, IL-10 and Ym1 in Aβ1–42-stimulated BV2 microglia [99]. In that study, TREM2 overexpression reduced the accumulation of Aβ and the expression of BACE-1, Iba-1, GFAP, TNF-α, IL-6 and IL-1β in both cortex and hippocampus in an APP/PS1 transgenic mouse model of AD [99]. In that study, TREM2 overexpression also led to the upregulation of M2 phenotype markers (Arg-1, IL-10 and Ym1) in both the cortex and hippocampus [99]. Interestingly, these effects were accompanied by the suppression of JAK2 and STAT1 activation and the reduction of SOCS1 and SOCS3 both in vivo and in vitro.
We also include new sentences as below.
Furthermore, recent observation showed that TREM2 upregulation reduces ApoE4-associated cognitive impairment and neurodegeneration [101]. These findings provide conflicting results showing whether the response of microglial TREM2 in AD pathophysiology is protective or detrimental. Considering on the results of preclinical studies, which show the beneficial effect of TREM2-associated activation of microglia on tau hyperphosphorylation [101,102] and Aβ phagocytosis [97,103], TREM2 could be a valuable therapeutic target in AD.
References)
[101] Mol Neurodegener 2020, 15, 57, doi:10.1186/s13024-020-00407-2.
[102] Nat Neurosci 2019, 22, 1217-1222, doi:10.1038/s41593-019-0433-0.
[103] Inflammation 2018, 41, 811-823, doi:10.1007/s10753-018-0735-5.
[104] Nat Neurosci 2019, 22, 191-204, doi:10.1038/s41593-018-0296-9.
The finding of a recent study by Park et al. showed that the GLP-1R agonist NLY01 inhibited the increased mRNA levels of TNF-α, IL-1α, IL-1β, IL-6 and C1q in Aβ-stimulated microglia, and also suppressed Aβ-induced MAP2, Bcl-2 and BDNF reduction and neuronal cell death [106].
→ The finding of a recent in vitro study by Park et al. showed that the GLP-1R agonist NLY01 inhibited the increased mRNA levels of TNF-α, IL-1α, IL-1β, IL-6 and C1q in Aβ-stimulated microglia, and also suppressed Aβ-induced MAP2, Bcl-2 and BDNF reduction and neuronal cell death [106].
- Microglia in Parkinson’s Disease (PD)
A recent observation from Wang et al. showed that the GLP-1R agonist PT320 ameliorated the progression of PD through improvements in behavior and DA midbrain function in a progressive PD mouse model [143]. Recently, the therapeutic use of GLP-1R agonists has been suggested in PD because of this receptor’s important role in the prevention of DA neuron loss and microglia-mediated neurotoxic A1 reactive astrocytes [104]. In that study, the GLP-1R agonist NLY01 effectively suppressed the α-synuclein preformed fibril (α-Syn PFF)-induced microglial activation marker Iba-1 and the secretion of TNF-α, IL-1α and C1q, and exerted inhibitory effects on the nuclear translocation of NF-κB in α-Syn PFF-stimulated microglia [104]. Therefore, these results suggest that the activation of GLP-1R in microglia could suppress A1 neurotoxic astrocytes and ameliorate progression in PD (Figure 2).
→ Recently, the therapeutic use of GLP-1R agonists has been suggested in PD because of this receptor’s important role in the prevention of DA neuron loss and microglia-mediated neurotoxic A1 reactive astrocytes [104]. In that study, the GLP-1R agonist NLY01 effectively suppressed the α-synuclein preformed fibril (α-Syn PFF)-induced microglial activation marker Iba-1 and the secretion of TNF-α, IL-1α and C1q, and exerted inhibitory effects on the nuclear translocation of NF-κB in α-Syn PFF-stimulated microglia [104]. A recent observation from Wang et al. showed that the GLP-1R agonist PT320 ameliorated the progression of PD through improvements in behavior and DA midbrain function in a progressive PD mouse model [143]. Therefore, these results suggest that the activation of GLP-1R in microglia could suppress A1 neurotoxic astrocytes and ameliorate progression in PD (Figure 2).
5 - In addition to the suggested addition in the conclusion, it would be interesting to have a paragraph analyzing/hypothesizing why some targets are found as "key" for different diseases, how this can be related with the role of microglia in the different diseases, its relation with possible therapeutic approaches, etc.
→ Thank you so much for your comments, which improve the quality of work. As your suggestion, we included sentences in Conclusion as below.
The identification of promising and shared therapeutic targets, such as DPP-4, GLP-1R and TrκB in the different diseases, such as AD, PD and HD suggest its importance in therapeutic approach in NDs.
6 - It would be also nice to have some lines explaining if some of the listed targets or molecules have been identified in an attempt to target modulate microglia to treat NDs.
→ Thank you so much for your comments, which improve the quality of work. As your suggestion, we included Table.1, which reflect the target of microglia in NDs. There are shared targets, such as DPP-4, GLP-1R and TrkB in NDs, such as AD, PD and HD. Thank you again for providing contribution to improving the quality of work.
Table 1. Microglia-targeting for NDs
|
Diseases |
Targeting |
|
Agent |
Reference |
|
Epilepsy |
GLP-1R |
Agonist |
Liraglutide |
[40] |
|
TRPV4 |
Antagonist |
HC-067047 |
[42] |
|
|
CX3CR1 |
Antibody |
Anti-CX3CR1 antibody |
[22] |
|
|
AD |
DPP-4 |
Inhibitor |
Saxagliptin |
[102] |
|
Linagliptin |
[112] |
|||
|
GLP-1R |
Agonist |
Exendin-4 |
[105] |
|
|
NLY01 |
[106] |
|||
|
TREM2 |
Agonist |
AL002c |
[100] |
|
|
TrкB |
Agonist |
AS86 |
[114] |
|
|
7,8-dihydroxyflavone |
[115] |
|||
|
PD |
DPP-4 |
Inhibitor |
Saxagliptin |
[139][142] |
|
Linagliptin |
[150] |
|||
|
GLP-1R |
Agonist |
NLY01 |
[104] |
|
|
PT320 |
[143] |
|||
|
TrкB |
Agonist |
7,8-dihydroxyflavone |
[152][153] |
|
|
HD |
DPP-4 |
Inhibitor |
Vildagliptin |
[156] |
|
GLP-1R |
Agonist |
Exendin-4 |
[142] |
|
|
TrкB |
Agonist |
7,8-dihydroxyflavone |
[156] |
Round 2
Reviewer 1 Report
The authors have addressed my comments properly
Reviewer 2 Report
The manuscript is now more complete and clear than before. The concerns of the reviewer have been addressed.